# Proficiency and Practices of Nursing Professionals in Meeting Patients’ Spiritual Needs within Palliative Care Services: A Nationwide, Cross-Sectional Study

**DOI:** 10.3390/healthcare12070725

**Published:** 2024-03-26

**Authors:** Tina Košanski, Marijana Neuberg

**Affiliations:** 1Faculty of Health Sciences, University of Novo mesto, 8000 Novo Mesto, Slovenia; mneuberg@unin.hr; 2Department of Nursing, University North, 42000 Varaždin, Croatia

**Keywords:** spirituality, spiritual needs, palliative care, nurses

## Abstract

Spirituality and spiritual needs are integral parts of the human experience, but they are often particularly important for palliative care patients. Spirituality has numerous positive effects, especially for those dealing with serious illness. Nevertheless, the spiritual dimension is sometimes overlooked in patient care. This study aims to determine the frequency of addressing the spiritual needs of palliative care patients in Croatia and to investigate the self-perceived confidence of caregivers in this task. A quantitative cross-sectional study was conducted involving 194 nurses in specialised palliative care services across Croatia. A specially developed and validated questionnaire was used for this study. The most common intervention undertaken by respondents was “promoting hope and optimism in patients” (88.4%), while the least common intervention was “reading books and other publications to patients” (13.9%). No statistically significant differences were found in the frequency of spiritual care in relation to the respondent’s level of education, professional experience and nursing environment. Approximately two-thirds of the surveyed nurses stated that they “often” or “always” provided some kind of spiritual care to palliative care patients. However, study participants who indicated that they had received sufficient formal instruction in addressing spiritual needs and spiritual care interventions demonstrated a statistically significant tendency to engage in these practices, as well as greater confidence in their knowledge and skills in this area compared to those who lacked such training. The study suggests that there is a need to identify existing barriers to the provision of spiritual care and to develop strategies to overcome them. By placing emphasis on the spiritual needs and preferences of patients, nursing professionals and other healthcare providers have the opportunity to elevate the standard of holistic care and foster a sense of comfort and dignity among patients.

## 1. Introduction

Due to the ageing of the population and the associated increase in chronic, i.e., noncommunicable diseases, the need for palliative care continues to grow, drawing attention to the importance of this area of biomedicine and healthcare. Studies indicate that 69–82% of people in high-income countries require palliative care [1]. Currently, the annual number of people needing palliative care is estimated at 56.8 million [2], with projections suggesting a 25.0% to 42.4% relative increase by 2040 (primarily driven by the ageing population) [3]. These statistics emphasise the importance of investing in the further development of palliative care, improving its quality and determining its current status in individual countries.

Palliative care is established in Croatia and is implemented on three levels:
At the first level, referred to as the “palliative approach”, palliative methods and procedures are integrated into settings that are not primarily focused on palliative care. Healthcare professionals at this level have basic knowledge of palliative care.The second level includes “general palliative care”, which is provided by healthcare professionals who frequently deal with patients requiring palliative care, even if this is not their primary area of responsibility, such as in oncology and neurology departments.

The third level comprises “specialised palliative care”, which deals with the complex needs of palliative care patients and entails advanced training, dedicated staff, and other resources. Specialised palliative care requires a collaborative interdisciplinary approach and a multiprofessional team [4,5]. In addition, all members of the specialised palliative care teams should have additional training in palliative care [6].

The forms of specialised palliative care in Croatia include various services, such as palliative care coordination centres, mobile palliative care teams, palliative care facilities (hospices), palliative care units, palliative care beds, palliative day hospitals, palliative care clinics, palliative care hospital teams, equipment rentals, volunteers, and civil society organisations [6].

All three levels of palliative care are available in the Croatian healthcare system, namely in primary healthcare (e.g., mobile palliative care teams, GP practices, community nursing services, home care, social welfare centres, and inpatient clinics in health centres), secondary healthcare (e.g., general and county hospital), and tertiary healthcare (e.g., clinics, clinical hospitals, and clinical hospital centres) [7].

An important concern identified both in the literature and in practice is that the care provided to palliative care patients is incomplete as it ignores their diverse and numerous needs and often does not take into account all dimensions of the individual [8,9,10]. More specifically, despite palliative care’s focus on providing holistic support—including physical, emotional, and social aspects—the spiritual dimension is frequently overlooked.

Spiritual needs in palliative care can be defined as an individual’s sense of peace, purpose, connection to others, and beliefs about the meaning of life [11,12]. The challenges related to spiritual well-being, saying goodbye to loved ones, maintaining hope and other end-of-life concerns are often poorly understood and not sufficiently integrated into patients’ overall care plans [11]. As a result, spiritual care and support (which is not necessarily synonymous with religion) are often labelled as the most overlooked dimension of patient care [12].

This study aims to determine the frequency of interventions performed by nurses in Croatia (such as active listening, engaging in discussions, compassionate presence, promoting spiritual reflection and reading, to name a few) in order to address the spiritual needs of palliative care patients and their self-perceived confidence in this task.

## 2. Materials and Methods

### 2.1. Research Design and Methodology of Data Collection

To collect the research data, a quantitative cross-sectional survey was conducted in Croatia from January to June 2023. The research was carried out as part of work towards a doctoral thesis to obtain relevant knowledge about the implementation of the holistic approach in palliative care, and the presented results are an integral part of the dissertation.

### 2.2. Measuring Instrument

A questionnaire was created for this study based on the relevant academic and professional literature [13,14,15,16,17,18].

The questionnaire consisted of several parts:

In the first part, socio-demographic data were collected (gender, age, years of professional experience in healthcare, years of professional experience in palliative care, level of education, and work setting).

The second part of the questionnaire collected data on the frequency of addressing patients’ spiritual needs, specifically the frequency of undertaking selected interventions. Participants were asked to indicate the frequency of performing each intervention on a 5-point Likert scale, where 1—never; 2—rarely; 3—sometimes; 4—often; 5—always.

In the third part of the questionnaire, respondents were asked to self-assess their confidence in the stated interventions on a 5-point Likert scale: 1—poor; 2—fair; 3—good; 4—very good; 5—excellent.

The fourth section of the questionnaire dealt with the importance of holistic care in the formal education of nurses. Participants were asked to indicate their level of agreement with the statements about the importance given to the spiritual dimension in nursing care during their education, i.e., whether they felt that patients’ spiritual needs were adequately addressed. Responses were rated on a 5-point Likert scale: 1—I strongly disagree; 2—I disagree; 3—I neither agree nor disagree; 4—I agree; 5—I strongly agree.

### 2.3. Sample

The sample is a random sample. As accurate records of nurses in specialised palliative care services were not available, the authors contacted all services with palliative care beds at the secondary and tertiary healthcare levels and healthcare centres in all counties in Croatia to determine the number of nurses in roles such as palliative care coordinators and members of mobile palliative care team, as well as those working in inpatient clinics with palliative care beds in health centres at the primary healthcare level. According to the available information, 313 nurses with different levels of education were working in specialised palliative care services throughout Croatia during the study period (personal communication, unpublished).

Our total sample included 194 nurses with different levels of education working in specialised palliative care services in Croatia (palliative care coordinators, members of mobile palliative care teams, nurses in inpatient clinics at health centres, nurses in palliative care facilities, and nurses in palliative care units). In Croatia, only nurses in specialised palliative care services exclusively care for patients needing palliative care, while nurses at other levels also care for other patients.

The criteria for inclusion in the study were as follows:Participants were general care nurses holding a bachelor’s, master’s, or doctorate degree in nursing.At the time of the survey, participants were working in specialised palliative care services in roles such as palliative care coordinators, members of mobile palliative care teams, nursing staff in inpatient clinics in health centres with palliative care beds, nursing staff in palliative care units, or nursing staff in palliative care facilities (hospices).Only palliative care coordinators who were also working in mobile palliative care teams (as stand-ins or because the team was understaffed) at the time of the survey were eligible for participation.

A total of 286 questionnaires were distributed in the workplace of nursing professionals, of which 194 correctly completed questionnaires (67.8%) and 1 incorrectly completed questionnaire were returned.

### 2.4. Ethical Considerations

General ethical approval for the research was obtained from the University of Novo mesto (FZV-282/2021). The ethics committees of all primary, secondary, and tertiary healthcare institutions in Croatia that provide specialised palliative care were sent research application forms. Of the total of 40 institutions that provide specialised palliative care (28 institutions at the primary, 11 institutions at the secondary, and 1 institution at the tertiary level of healthcare), consent was obtained from 33 institutions (23 institutions at the primary level of healthcare, 9 institutions at the secondary level of healthcare, and 1 institution at the tertiary level of healthcare). Before completing the survey, participants were informed about the purpose of the study, the use of the data, the assurance of anonymity, risk-free participation, and the possibility of voluntary withdrawal, which they confirmed by signing the consent form.

### 2.5. Statistical Analysis of the Data

The data obtained from the questionnaire were presented descriptively and in tabular form. The descriptions and tables were created using Microsoft Office Word, while IBM SPSS Statistics Version 29.0.1 was used for the statistical analysis. Numerical data were tested for normal distribution using the Smirnov–Kolmogorov test, and appropriate parametric statistical tests were applied depending on the results.

Quantitative data were presented as means, standard deviations, and 95% confidence intervals, while categorical data were presented as absolute frequencies and proportions or 95% confidence intervals.

The correlation between the work setting and the frequency of interventions by nurses/technicians in specialised palliative care services was analysed using the one-way ANOVA. Pearson’s correlation coefficient (r) was used to analyse the relationship between frequency of interventions, confidence level, years of experience, level of education, and inclusion of holistic care in nursing education among nurses/technicians in specialised palliative care services.

A *p*-value of less than 0.05 was considered statistically significant (two-tailed).

## 3. Results

### 3.1. Socio-Demographic Categorical and Quantitative Variables

Table 1 shows the descriptive statistics of the socio-demographic categorical variables for the entire sample (N = 194). The majority of the sample analysed are women (177, or 91.20%). Most of the respondents, 96 of them (49.5%), have completed training for general care nurses. Regarding the organisation of working hours, 12 h tours were the most common (94 or 48.5% of respondents), followed by early shift work (85 or 29.90% of respondents). For almost half of the respondents (92 or 47.40%), the functional model of healthcare organisation was the predominant model. In terms of the work setting, most respondents (109 or 56.20%) worked in a palliative care facility/unit, while 38 respondents (19.60%) were members of mobile palliative care teams. In terms of patient categorisation, 138 (71.10%) of the respondents most commonly cared for patients who fell into category IV.

Table 2 shows the descriptive statistics of the socio-demographic quantitative variables for the entire sample. The mean age (SD) of the respondents is 38.55 (12.08) years. The average professional experience in healthcare is 16.84 (11.79) years, while the average professional experience in palliative care is 5.24 (4.71) years.

### 3.2. Internal Consistency Analysis for the Subscale “Frequency of Addressing Patients’ Spiritual Needs” and the Subscale “Confidence in Addressing Patients’ Spiritual Needs”

Table 3 shows the analysis of the internal consistency of spiritual care interventions. The Cronbach’s alpha coefficient for the indicated subscale is 0.777.

Table 4 shows an analysis of the internal consistency of the statements on the participants’ confidence in the spiritual care interventions listed. The Cronbach’s alpha coefficient for the indicated subscale is 0.832. The indicated subscale demonstrates a high level of internal consistency, suggesting that the items within this subscale reliably measure participants’ confidence.

### 3.3. Descriptive Data on the Frequency of Spiritual Care Interventions

Table 5 shows the respondents’ answers regarding the frequency with which the stated interventions were carried out. In total, 60.80% of respondents reported “always” promoting hope and optimism in patients. Only 15.50% said they “always” prayed with patients, and only 30.90% “always” ensured that patients had the opportunity to attend religious services. Reading books or other publications to patients is the least frequently performed intervention—22.70% of respondents “never” did so, while as many as 32.00% “rarely” did so.

### 3.4. Descriptive Data on Confidence Level in Spiritual Care Interventions

Table 6 shows the respondents’ self-reported level of confidence in the spiritual care interventions listed. In total, 41.20% of respondents indicated that their confidence in promoting hope and optimism in patients was “excellent”. In total, 34.00% stated that their confidence in praying with patients was “good”. The majority of respondents, 35.10%, considered their confidence in making sure that patients had the opportunity to attend religious services to be “good”. Similarly, the majority of respondents, 40.20%, rated their confidence in reading books and other publications as “good”. In total, 35.10% of respondents rated their confidence in talking to patients about their impending death as “very good”.

### 3.5. Correlation between Work Setting, Professional Experience, and Level of Education and the Frequency of Providing Spiritual Care and the Level of Confidence in Spiritual Care Interventions

The correlation between work setting and the frequency of spiritual care delivery and confidence in delivering these interventions among nurses/technicians in specialised palliative care services is shown in Table 7 and Table 8. ANOVA showed no significant differences.

The correlation between professional experience in healthcare, professional experience in palliative care, level of education, the importance given to spiritual care in nursing education, and the frequency of spiritual care and level of confidence in spiritual care is shown in Table 9. A statistically significant correlation was determined using Pearson’s correlation coefficient.

Years of work experience in the healthcare system and in palliative care, as well as the level of education, did not demonstrate statistically significant correlations with the frequency of implementing interventions for addressing spiritual needs or with knowledge about them. However, participants who received sufficient instruction on spiritual care during their formal education statistically significantly more frequently engage in spiritual care and possess greater knowledge about it.

## 4. Discussion

Palliative care involves the active and holistic care of people with advanced, progressive illnesses that do not respond to curative treatment. It includes the relief of pain and other distressing symptoms as well as the provision of psychological, social, and spiritual support for the patient and their family [19,20,21], respecting their values [22]. Palliative care takes an interdisciplinary approach in which nurses, alongside other professionals, play a key role in maintaining quality of life and supporting patients and their families [23].

By definition, palliative care involves comprehensive, i.e., holistic care. This holistic approach is essential not only for patients diagnosed with chronic illnesses but also for those receiving palliative care and should form the core of daily care provided to critically ill patients [24]. Holistic nursing is defined as a nursing practice that aims to address the entirety of the individual, taking into consideration the physical, mental, social, and spiritual dimensions of the patient [25].

All dimensions of a person must be taken into account in patient care; however, in palliative care patients, spiritual needs often take centre stage [26,27]. Spirituality is expressed in values, beliefs, and traditions and involves the pursuit and expression of meaning and purpose as well as a sense of connectedness to oneself, others, nature, or something that is meaningful or sacred to the individual [28]. Spirituality is a deeply personal experience and is widely recognised as beneficial in dealing with challenging situations such as terminal illness [29,30].

A survey involving 194 nurses working in specialised palliative care services in Croatia examined the frequency of interventions that address the spiritual needs of palliative care patients and the nurses’ self-reported level of confidence in these interventions. The descriptive data reveal that among the interventions listed, promoting hope and optimism in patients was the most common intervention. It was reported to have been undertaken by the majority of interrogated nurses, with most of them assessing their confidence in performing this activity as “very good” or “excellent”. Just over half of the participants reported frequently engaging in discussions with patients about their imminent death when patients felt the need to do so, with most assessing their confidence in performing this activity as “very good” or “excellent”. Moreover, almost two-thirds of study participants stated that they frequently discussed the meaning of life and past accomplishments with their patients, with most stating that their confidence in doing so was “very good” or “excellent”. However, only 13.9% of respondents reported frequently reading books, newspapers, and other publications to patients when needed, with a little over one-third of them rating their level of confidence in this activity as “very good” or “excellent”. More than two-thirds of respondents admitted to “never” or “rarely” praying with patients or doing so “sometimes”; furthermore, only half of the study participants reported making sure that patients “always” or “very often” have the opportunity to attend religious services, with 48.5% expressing “very good” or “excellent” confidence in this regard. Interestingly, more than half of respondents either “always” or “very often” involved a priest or other religious official in patient care, with 57.2% feeling “very good” or “excellent” in their confidence levels for this intervention. The authors Burkhardt and Nagail-Jacobson point out that spirituality is the least understood human dimension in nursing practice and is often misunderstood as purely religious needs [31]. This study found that more than half of respondents engage in interventions that include various aspects of spirituality beyond religiosity. However, the intervention “reading books and other publications to patients” is only undertaken by a small percentage of participants (just over 10%), despite research demonstrating its benefits to patient well-being [32,33,34]. A systematic literature review demonstrates how nurses highlight shortcomings in their initial nursing education, expressing a need for additional training, comprehension, and professional support to recognise and deliver spiritual care effectively [35].

For patients who are religious and fulfil their spiritual needs through religious rites and rituals, the nurse should act as an intermediary between the patient and a priest, who, unlike the nurse, may not always be available to the patient [36]. However, in the present study, it was found that more than 60% of participants did not pray with patients, and only slightly more than half of them made sure that patients had the opportunity to attend religious services or involved a priest or other religious official in patient care. A study conducted in Saudi Arabia demonstrated that religious practices such as prayer and Quran recitation can contribute to an overall improvement in health and help cancer patients better cope with their illness [37]. Moreover, higher levels of spirituality have been associated with fewer depressive disorders [38], which are common in palliative care patients [39], indicating the importance of supporting patients to engage in spiritual practices. A study on spiritual care practices in nursing care revealed that 93% of nurses recognised that patients have spiritual needs, but only about two-thirds of participants reported actively engaging in spiritual care practices [40], which is consistent with the findings of this study. For religious patients, religion is the most important dimension of spirituality [38], indicating the need to improve spiritual care for religious patients. O’Callaghan et al. conclude that spiritual care providers in hospitals should strive to recognise individuals who are in need of pastoral or religiously oriented support [41]. It is noteworthy that patients and their families generally expect empathy and compassion from caregivers rather than profound spiritual care [27], a factor to be considered in daily practice.

No statistically significant differences were observed in the frequency of spiritual care interventions in relation to work settings in specialised palliative care services. Likewise, there were no statistically significant differences in confidence in interventions that address the spiritual needs of palliative care patients. These results are not surprising given that nurses’ education and additional training are not dependent on their work setting in specialised palliative care services. Furthermore, professional experience in healthcare, palliative care, and level of education had no statistically significant impact on the frequency of spiritual care interventions or confidence in delivering these interventions. This result was surprising, as one would expect that nurses with longer work experience and higher levels of education would have greater confidence in their knowledge and skills in spiritual care interventions and would perform them more frequently due to formal and experiential learning, as Filej and Kaučić found in their study on holistic care [42]. This suggests the possibility that the formal training of nurses may have neglected the spiritual needs of patients and the interventions to address them. However, participants who reported receiving adequate formal training in spiritual needs and spiritual care interventions were statistically significantly more likely to perform them and reported higher confidence in their knowledge and skills in this area compared to those who did not. This suggests that factors other than training influence the delivery of spiritual care to patients. Mugia et al. point out that nurses may not provide spiritual care to patients in their daily work due to a lack of time or skills [43]. Balboni et al. claim that failure to adequately address patients’ spiritual needs primarily results from inadequate staff training, with nurses often advocating for additional training in this aspect of care [44]. Selman et al. maintain that all healthcare professionals, particularly in palliative care, should have a basic knowledge of spiritual care [45]. In addition to knowledge gaps, other authors cite increased workload, shortage of nursing staff, and lack of guidelines for spiritual care as reasons for neglecting patients’ spiritual needs [46,47,48]. Considering that nurses in Croatia experience similar challenges and barriers, these factors may contribute to the fact that patients’ spiritual needs are not sufficiently addressed in that country as well. Future research should, therefore, focus on identifying barriers to the provision of spiritual care to patients.

In the context of palliative care, understanding and addressing religious and spiritual struggles are crucial aspects of providing holistic care to patients facing end-of-life challenges [49,50]. These struggles can profoundly impact patients’ emotional well-being, coping mechanisms, and overall quality of life [49,50]. Certainly, we also have to acknowledge potential differences in religious or spiritual background between caregiver and patient that could play a role in the desired and provided spiritual care. Additionally, there is the matter of a nonreligious nurse offering prayers to a religious patient, which warrants further exploration of how it impacts the overall process of care.

There are several limitations to our study. Although we utilized a random sample of nurses from specialized palliative care services in Croatia, it is important to note that the sample may not be fully representative of all nurses working in palliative care in the Republic of Croatia. More specifically, biases in the selection process (such as certain institutions being overrepresented or underrepresented) could have potentially impacted the generalizability of our findings. There is also a possibility of response bias, and self-assessments of confidence levels may not always align with actual performance. Given the sensitive nature of the topics covered in the questionnaire, such as addressing patients’ spiritual needs and the importance of holistic care, respondents might have felt pressured to provide responses that align with societal or professional expectations rather than expressing their true opinions or practices. We also did not extensively explore potential confounding variables that could influence the frequency of addressing spiritual needs or confidence levels, such as individual religious beliefs or cultural backgrounds of the nurses, variations in patient demographics, and preferences or organizational factors within healthcare settings. Finally, there may be other important aspects of holistic care or spiritual support that were not included in the questionnaire, potentially limiting the comprehensiveness of the findings.

## 5. Conclusions

This study shows that approximately two-thirds of nurses “often” or “always” provide some kind of spiritual care to palliative care patients. It suggests the need to identify and remove barriers to the provision of spiritual care, whether through improving nursing education or through organisational changes in specialised palliative care services in Croatia. There is also a need for improved collaboration in healthcare. While nurses bear the primary responsibility for patient care, involving other healthcare professionals (such as clergy members) can enrich the spiritual support provided to patients, which definitely aligns with the holistic nature of palliative care and promotes comprehensive well-being. Although existing frameworks offer valuable insights into holistic patient care, dedicated guidelines focusing on spiritual support should be developed to provide nursing professionals with clear direction and standardised approaches. Finally, our study prompts reflection on healthcare practices in general, emphasising the importance of patient-centred care. By prioritising patients’ spiritual needs and preferences, healthcare providers can enhance the quality of care and promote patients’ comfort and dignity.

## Figures and Tables

**Table 1 healthcare-12-00725-t001:** Descriptive statistics of socio-demographic categorical variables for the entire sample (N = 194).

		N	%	95% CI	
Gender	Male	17	8.80%	5.40%	13.40%
	Female	177	91.20%	86.60%	94.60%
Level of education	General care nurse/technician	96	49.50%	42.50%	56.50%
	Undergraduate professional/university study in nursing	62	32.00%	25.70%	38.80%
	Graduate professional/university study in nursing	33	17.00%	12.20%	22.80%
	Doctoral study programme	3	1.50%	0.40%	4.10%
Organisation of working hours	Early shift	58	29.90%	23.80%	36.60%
	Early and afternoon shifts	17	8.80%	5.40%	13.40%
	Early, afternoon and night shifts	24	12.40%	8.30%	17.60%
	12 h tours	94	48.50%	41.50%	55.50%
	Shift work with on-call duty	1	0.50%	0.10%	2.40%
Organisational model	Functional	92	47.40%	40.50%	54.40%
	Team	61	31.40%	25.20%	38.20%
	Primary	41	21.10%	15.80%	27.30%
Work setting	Member of a mobile palliative care team	38	19.60%	14.50%	25.60%
	Palliative care coordinator	15	7.70%	4.60%	12.10%
	Nurse/technician in an inpatient clinic at the health centre	23	11.90%	7.90%	17.00%
	Nurse/technician in a palliative care facility/unit	109	56.20%	49.20%	63.00%
	Nurse/technician in a hospice	9	4.60%	2.30%	8.30%

**Table 2 healthcare-12-00725-t002:** Descriptive statistics of the socio-demographic quantitative variables for the entire sample (N = 194).

	Arithmetic Mean	SD	Min	Max	Centile		
					25	Median	75
Age	38.55	12.08	20.00	65.00	29.00	37.50	47.25
Years of experience in healthcare	16.84	11.79	1.00	45.00	7.00	15.00	25.00
Years of experience in palliative care	5.24	4.71	0.20	39.00	2.00	4.00	7.00

**Table 3 healthcare-12-00725-t003:** Analysis of the internal consistency of spiritual care interventions.

Cronbach’s Alpha Coefficient	Number of Items	
0.777	7	
	Arithmetic mean	SD
I promote hope and optimism in patients.	4.49	0.707
I pray with patients if needed.	2.86	1.315
I make sure that patients have the opportunity to attend religious services.	3.56	1.213
I read books, newspapers, and other publications to patients when needed.	2.44	1.128
I involve a member of clergy in patient care.	3.48	1.26
I engage in discussions with patients about the meaning of life and their past accomplishments.	3.85	0.964
I talk to patients about their imminent death if they feel the need to do so.	3.67	1.055
	Corrected item correlation-total score	Cronbach’s alpha coefficient if item deleted
I promote hope and optimism in patients.	0.384	0.772
I pray with patients if needed.	0.559	0.738
I make sure that patients have the opportunity to attend religious services.	0.57	0.735
I read books, newspapers, and other publications to patients when needed.	0.556	0.738
I involve a member of clergy in patient care.	0.595	0.729
I engage in discussions with patients about the meaning of life and their past accomplishments.	0.404	0.767
I talk to patients about their imminent death if they feel the need to do so.	0.444	0.76

**Table 4 healthcare-12-00725-t004:** Analysis of the internal consistency of the statements on confidence in spiritual care interventions.

Cronbach’s Alpha Coefficient	Number of Items	
0.832	7	
	Arithmetic mean	SD
Promoting hope and optimism in patients.	4.18	0.823
Praying with patients.	3.32	1.148
Making sure that patients have the opportunity to attend religious services.	3.46	1.097
Reading books, newspapers, and other publications to patients.	3.26	1.095
Involving a member of clergy in patient care.	3.68	1.116
Engaging in discussions with patients about the meaning of life and their past accomplishments.	3.94	0.894
Talking to patients about their impending death.	3.72	1
	Corrected item correlation—total score	Cronbach’s alfa coefficient if item deleted
Promoting hope and optimism in patients.	0.487	0.823
Praying with patients.	0.657	0.796
Making sure that patients have the opportunity to attend religious services.	0.699	0.789
Reading books, newspapers and other publications to patients.	0.562	0.813
Involving a member of clergy in patient care.	0.569	0.811
Engaging in discussions with patients about the meaning of life and their past accomplishments.	0.577	0.811
Talking to patients about their impending death.	0.518	0.819

**Table 5 healthcare-12-00725-t005:** Frequency of responses regarding the frequency of the indicated spiritual care interventions (N = 194).

Intervention		N	%	95% CI	
I promote hope and optimism in patients.	Never	0	0.00%		
	Rarely	1	0.50%	0.10%	2.40%
	Sometimes	21	10.80%	7.00%	15.80%
	Very often	54	27.80%	21.90%	34.40%
	Always	118	60.80%	53.80%	67.50%
I pray with patients if needed.	Never	37	19.10%	14.00%	25.00%
	Rarely	42	21.60%	16.30%	27.80%
	Sometimes	57	29.40%	23.30%	36.10%
	Very often	28	14.40%	10.00%	19.90%
	Always	30	15.50%	10.90%	21.00%
I make sure that patients have the opportunity to attend religious services.	Never	8	4.10%	2.00%	7.60%
	Rarely	34	17.50%	12.70%	23.30%
	Sometimes	54	27.80%	21.90%	34.40%
	Very often	38	19.60%	14.50%	25.60%
	Always	60	30.90%	24.70%	37.70%
I read books, newspapers and other publications to patients when needed.	Never	44	22.70%	17.20%	28.90%
	Rarely	62	32.00%	25.70%	38.80%
	Sometimes	61	31.40%	25.20%	38.20%
	Very often	13	6.70%	3.80%	10.90%
	Always	14	7.20%	4.20%	11.50%
I involve a member of clergy in patient care.	Never	19	9.80%	6.20%	14.60%
	Rarely	23	11.90%	7.90%	17.00%
	Sometimes	47	24.20%	18.60%	30.60%
	Very often	56	28.90%	22.80%	35.50%
	Always	49	25.30%	19.50%	31.70%
I engage in discussions with patients about the meaning of life and their past accomplishments.	Never	3	1.50%	0.40%	4.10%
	Rarely	10	5.20%	2.70%	9.00%
	Sometimes	59	30.40%	24.30%	37.10%
	Very often	64	33.00%	26.70%	39.80%
	Always	58	29.90%	23.80%	36.60%
I talk to patients about their impending death if they feel the need to do so.	Never	5	2.60%	1.00%	5.60%
	Rarely	16	8.20%	5.00%	12.70%
	Sometimes	72	37.10%	30.50%	44.10%
	Very often	46	23.70%	18.10%	30.10%
	Always	55	28.40%	22.40%	35.00%

**Table 6 healthcare-12-00725-t006:** Frequency of responses on the self-reported confidence level in the listed spiritual care interventions (N = 194).

Confidence Level in the Following Areas		N	%	95% CI	
Promoting hope and optimism in patients	Poor	1	0.50%	0.10%	2.40%
	Fair	3	1.50%	0.40%	4.10%
	Good	36	18.60%	13.60%	24.50%
	Very good	74	38.10%	31.50%	45.10%
	Excellent	80	41.20%	34.50%	48.30%
Praying with patients	Poor	17	8.80%	5.40%	13.40%
	Fair	23	11.90%	7.90%	17.00%
	Good	66	34.00%	27.60%	40.90%
	Very good	56	28.90%	22.80%	35.50%
	Excellent	32	16.50%	11.80%	22.20%
Making sure that patients have the opportunity to attend religious services	Poor	11	5.70%	3.00%	9.60%
	Fair	21	10.80%	7.00%	15.80%
	Good	68	35.10%	28.60%	41.90%
	Very good	56	28.90%	22.80%	35.50%
	Excellent	38	19.60%	14.50%	25.60%
Reading books, newspapers, and other publications to patients	Poor	14	7.20%	4.20%	11.50%
	Fair	26	13.40%	9.20%	18.70%
	Good	78	40.20%	33.50%	47.20%
	Very good	47	24.20%	18.60%	30.60%
	Excellent	29	14.90%	10.50%	20.50%
Involving a member of clergy in patient care	Poor	8	4.10%	2.00%	7.60%
	Fair	24	12.40%	8.30%	17.60%
	Good	41	21.10%	15.80%	27.30%
	Very good	70	36.10%	29.60%	43.00%
	Excellent	51	26.30%	20.50%	32.80%
Engaging in discussions with patients about the meaning of life and their past accomplishments	Poor	1	0.50%	0.10%	2.40%
	Fair	11	5.70%	3.00%	9.60%
	Good	44	22.70%	17.20%	28.90%
	Very good	80	41.20%	34.50%	48.30%
	Excellent	58	29.90%	23.80%	36.60%
Talking to patients about their impending death	Poor	2	1.00%	0.20%	3.30%
	Fair	22	11.30%	7.50%	16.40%
	Good	53	27.30%	21.40%	33.90%
	Very good	68	35.10%	28.60%	41.90%
	Excellent	49	25.30%	19.50%	31.70%

**Table 7 healthcare-12-00725-t007:** Correlation between work setting and the frequency of spiritual care and the level of confidence in spiritual care interventions.

		N	Arithmetic Mean	SD	95% CI	
					Lower	Upper
Frequency of spiritual care	Member of a mobile palliative care team	38	3.27	0.77	3.02	3.53
	Palliative care coordinator	15	3.60	0.59	3.27	3.93
	Nurse/technician in an inpatient clinic at the health centre	23	3.71	0.66	3.43	4.00
	Nurse technician at a palliative care facility/unit	109	3.49	0.73	3.35	3.63
	Nurse/technician at a hospice	9	3.40	0.73	2.83	3.96
Confidence in spiritual care	Member of a mobile palliative care team	38	3.64	0.66	3.43	3.86
	Palliative care coordinator	15	4.10	0.64	3.74	4.45
	Nurse/technician in an inpatient clinic at a health centre	23	3.79	0.65	3.51	4.07
	Nurse technician at a palliative care facility/unit	109	3.58	0.76	3.44	3.73
	Nurse/technician at a hospice	9	3.46	0.68	2.94	3.98

**Table 8 healthcare-12-00725-t008:** Results of the one-way ANOVA in relation to the data from Table 7.

		Sum of Squares	df	Mean Squares	F	P
Frequency of spiritual care	Between groups	3.148	4	0.787	1.511	0.201
	Within groups	98.428	189	0.521		
	Total	101.576	193			
Confidence in spiritual care	Between groups	4.227	4	1.057	2.039	0.091
	Within groups	97.967	189	0.518		
	Total	102.194	193			

**Table 9 healthcare-12-00725-t009:** Correlation between professional experience in healthcare, professional experience in palliative care, level of education, importance given to spiritual care in nursing education, and the frequency of spiritual care and the level of confidence in spiritual care.

		Spiritual Care Interventions	Confidence in Spiritual Care
Experience in healthcare	r	−0.009	<0.001
	P	0.851	0.999
Experience in palliative care	r	0.085	0.055
	P	0.103	0.294
Level of education	r	−0.017	0.062
	P	0.772	0.280
Sufficient importance given to spiritual care in nursing in formal education	r	0.125	0.297
	P	0.022	<0.001

## Data Availability

The data presented in this study are available from the corresponding author on request.

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
