# Peer review of "Proficiency and Practices of Nursing Professionals in Meeting Patients’ Spiritual Needs within Palliative Care Services: A Nationwide, Cross-Sectional Study"

_healthcare, 2024, doi:10.3390/healthcare12070725_

Round 1
Reviewer 1 Report
Comments and Suggestions for Authors
In general, an engaging and clearly formulated paper, for which compliments are due! However, there are still some points where the article could be improved and clarified, or expanded with additional perspectives.
Abstract
Line 12/13 Nevertheless – care” à "If I look at the results, it doesn't seem too bad? Or is it about international research?"
Line 26 “Also” à why also?
Introduction
Line 37 That looks rather peculiar, because it's about once in a lifetime and not the percentage itself I guess? The same applies to the increase mentioned, I think that's relative and not absolute, otherwise you'd go above 100.
Line 50 The dot is missing.
General
It's good that attention is being paid to Croatia and how palliative care is
being implemented there. However, the transition to missing the spiritual
dimension seems a bit abrupt. What does palliative care entail? Is there only
focus on physical aspects in current care? And why is spiritual care
important? Are there examples where this is done well or initiatives to
improve this? Additionally, the next paragraph suddenly shifts to
interventions. What interventions are known or recommended?
Materials & Method
Line 89 Talking about 'the' but they have not been mentioned yet.
Measuring instrument: I wonder: did you not collect data on respondents won religious or spiritual background? This is known to be a predictor of spiritual care. If not, the matter could be discussed.
Sample: How where potential participants approached?
Line 106-107-108 Why the research aim here?
Line 111 etc. I see some relevant data collection information is provided here. I would suggest to put this before the information in the first paragraph and to avoid repeating information from the introduction.
Line 140 33 institutions: is that the reason why not 313 questionnaires but 286 were distributed?
Line 153-154 Why 95% confidence intervals from categorical and dichotomous data??
Results
The results are very, very extensive and contain a lot of tables. I'm not exactly
sure what the requirements of the journal are in this regard, but I would
recommend shortening and clustering more. Some data could possibly be
summarized in the text. The 95% confidence intervals for dichotomous and
categorical variables remain a mystery to me. Why do you present those?
Have you also tested the assumptions to test Pearson correlations?
Discussion
Regarding the discussion, I have some questions and remarks. I wonder if
differences in religious or spiritual background between caregiver and
patient could still play a role in the desired and provided spiritual care. I
wouldn't repeat percentages in the discussion; that's distracting and they
have already been mentioned earlier. Of course, you can describe the main
points. Furthermore, I miss any reference to religious or spiritual struggles.
Can the authors discuss whether this could be relevant for the current study?
When it comes to prayer: can a non-religious nurse pray with a religious
patient?
Conclusion
364-365 Can you state this? It appears to be measured with different components? Perhaps one always prays while the other always provides hope. And another question: is it desirable for 100% of nurses to provide spiritual care? That may be so, but might be articulated then.
Reviewer 2 Report
Comments and Suggestions for Authors This is a paper on the provision of spiritual care and nurses' confidence in it. They examines the internal consistency of the scale they created and examines its correlation with nurses' experiences and work environment. The results are clearly stated. Although the results of descriptive statistics is a little long, it is thought to be of interest to Healthcare readers. Major comment Is it not necessary to compare the created scale with existing evaluation scales for spiritual care? Minor comment What does the patient category in Table 2 refer to? ECOG-PS? Cancer stage?Author Response
Please see the attachment.

Reviewer 3 Report
Comments and Suggestions for Authors
This study reports the results of a survey on palliative care specialist nurse attitude on patient’ spiritual needs support.
The study involves a large number of respondents, in different palliative care institution oin Croazia.
The major limit of the study, in my opinion, is the lack of a clear definition of the concept of spiritual needs
Previous paper report a description of spiritual needs in palliative care as an individual's sense of peace, purpose, and connection to others, and beliefs about the meaning of life. The term spirituality has not the same meaning of religion.
Some of the activities reported in the questionnaires are not specifically related to spiritual need, for example: to promote hope and optimism or to read paper or book, do not seem to me correspond to spiritual assistance.
Minor revisions
The same results are reported in the text and in several tables.
Round 2
Reviewer 3 Report
Comments and Suggestions for Authors
the revised version of the manuscript has been mildly improved but still remain the most important issues.
In my opinion the questionnaire utilized is very confusing and is not clearly focused on spiritual needs.
The topic is however of interest
